

# Face recognition is similarly affected by viewpoint in school-aged children and adults

Marisa Nordt and Sarah Weigelt

Department of Developmental Neuropsychology, Institute of Psychology, Ruhr-Universität Bochum, Bochum, Germany

## ABSTRACT

There is an ongoing debate on the question when face processing abilities mature. One aspect that has been part of this debate is the ability to recognize faces in and across different viewpoints. Here, we tested 128 participants consisting of school-age children (ages, 5 –10 years) and adults (ages, 19–37 years) in two experiments to investigate the effects of different viewpoints (including front, three-quarter, profile view) on face recognition during development. Furthermore, we compared recognition performance for faces to that of another object category (cars). In the first experiment ($n = 88$) we tested if the pattern of performance for faces presented in different viewpoints is similar in school-aged children and adults. Participants completed a two-alternative-forced-choice (2AFC) memory task comprising images of both faces and cars in front, three-quarter and profile view, which were presented in the same viewpoint during learning and testing. In the second experiment ($n = 40$) we tested if face recognition is similarly affected by viewpoint changes in children and adults. In this experiment the 2AFC memory task included a change of viewpoint between learning and testing. While in both experiments we found higher recognition performance for faces with increasing age, the overall pattern of both viewpoint and viewpoint-change-effects and also the difference between view-change- and no-change-conditions was similar across age groups. In contrast to faces, no viewpoint effects were observed in cars (experiment 1), viewpoint change effects, however, were similar for cars and faces (experiment 2). In sum, our results suggest early maturity of the ability to recognize faces in and across different viewpoints.

## INTRODUCTION

Even shortly after birth, newborns can distinguish their mothers face from the face of a stranger (*Bushnell, Sai & Mullin, 1989*), thereby demonstrating that basic face recognition abilities are present at such an early point in life. Furthermore, it has been shown that during the first year of life rapid changes with regard to face recognition abilities take place (*Pascalis, De Haan & Nelson, 2002*; *Turati, Bulf & Simion, 2008*).

Nevertheless, there are also studies that support a prolonged development of face recognition (*Lawrence et al., 2008*; *Germine, Duchaine & Nakayama, 2011*; *Weigelt et al., 2014*). Studies investigating unfamiliar face recognition from childhood to adulthood

Corresponding author
Marisa Nordt, marisa.nordt@rub.de

showed improvement of face recognition abilities between six and 10 years, followed by a short plateau between the ages of 10 and 13 (*Carey, Diamond & Woods, 1980*; *Lawrence et al., 2008*). *Germine, Duchaine & Nakayama (2011)* tested over 44,000 participants aged 10–70 demonstrating that the ability to recognize faces increases even further, reaching its peak level around 31 years. In contrast, performance levels for inverted faces peaked much earlier, at 23 years.

The question of early versus late maturity of face processing abilities has launched a debate in face processing research. Early studies suggested that most, if not all core processes of face recognition, like configural processing are not qualitatively present until age 10 years, thereby promoting a late maturity view (*Carey & Diamond, 1977*). However, when studies showed that important processes of face recognition are present early, but performance does not reach adult levels until adolescence, the late maturity view was modified to a late quantitative maturity view (*Maurer, Grand & Mondloch, 2002*). As an alternative explanation, the early maturity hypothesis has been proposed. It states that core aspects of face processing are present early, approximately at the age of five years and, more importantly, are at adult levels at this point (*Crookes & McKone, 2009*).

This debate also encompasses the development of the ability to detect and recognize faces in and across different viewpoints. This ability is of vital importance for our lives, as the viewpoint a face is seen from changes from encounter to encounter with a person in everyday life. A main finding from studies in adults on the effects of viewpoints is superior face recognition performance for three-quarter views compared to front and profile views of faces (*Bruce, Valentine & Baddeley, 1987*; *O'Toole, 1998*; *McKone, 2008*). A further viewpoint effect is lower recognition performance for profile views of faces, which is reflected in both longer reaction times and lower accuracy (e.g., *Bruce, Valentine & Baddeley, 1987*; *McKone, 2008*). It has been suggested that this decrement of performance in comparison to front and three-quarter views results from a lack of information on the configuration of internal features, like the spacing of the eyes (*Hill, Schyns & Akamatsu, 1997*), and from a rarer occurrence of this view (*McKone, 2008*).

Several studies have addressed the development of face recognition across different viewpoints during infancy. In a study by *Turati, Bulf & Simion (2008)*, one- to three-day old infants were habituated with a face in either front, three-quarter or profile view. In a next step, infants were tested with two faces comprising the target face in a novel view (also either front, three-quarter or profile) paired with a novel face presented in the same view. Turati and colleagues found a significant novelty preference for faces despite viewpoint changes between front and three-quarter view, however when profiles were included in either study or test phase no novelty preference was observed. This result is coherent with studies showing that the ability to encode faces in profile view only emerges at seven months (*Fagan III, 1976*). Further support for this finding comes from a near-infrared spectroscopy (NIRS) study in five- and eight-months-old infants (*Nakato et al., 2009*). Results of this study showed that in eight-months-old infants the concentration of oxy-hemoglobin (oxy-Hb) and of total Hb in right temporal brain regions increased for front and profile views of faces (relative to a condition in which other objects were shown), while for the younger age group concentration of oxy-Hb and total Hb increased for front views of

faces only. Thereby, these results suggest that eight-month-old but not five-month-old infants showed face-like processing of profile views of faces in right temporal brain regions. Together, behavioral as well as neuroimaging studies provide evidence that encoding of profile views of faces is the only face processing ability undergoing qualitative changes during infancy (for a review, see *McKone et al., 2012*).

Studies targeting the effects of viewpoint changes during childhood so far have focused on changes between front and three-quarter view (*Mondloch et al., 2003*; *Jeffery et al., 2013*; *Anzures et al., 2014*; *Crookes & Robbins, 2014*). *Mondloch et al. (2003)* tested six-, eight-, and 10-year-old children and adults on five different tasks, in which participants had to match either a certain identity despite changes in facial expression or changes in viewpoint, or they had to match a certain direction of eye gaze, a facial expression or had to choose which faces were mouthing certain vowels. While six-year-olds' performance was lower compared to adults on each of these tasks, 10-year-olds' accuracy differed from that of adults only in one task: matching of identities despite changes in viewpoint, thereby suggesting a late development of this ability. In contrast, *Crookes & Robbins (2014)* found no indication for a developmental change of face recognition across viewpoint change. They tested eight-year-old children and adults on a two-alternative forced-choice (2AFC) memory task including two conditions. In one condition there was no change of viewpoint between study and test phase (same-view condition): faces were presented in front view in both study and test phase. In the second condition there was a viewpoint change between study and test phase (change-view condition): faces were presented in front view in the study face and in three-quarter view in the test phase. Crookes and Robbins found similar results for adults and eight-year-olds, with higher accuracy in the same-view condition compared to the change-view condition in both groups, but showing neither a main effect of age group nor an interaction between condition and age group, thus indicating early maturity of face recognition across viewpoint changes. Likewise, *Jeffery et al. (2013)* showed that the transfer of face aftereffects across changes in viewpoint was comparable in seven- to nine-year-old children and adults. Face aftereffects refer to the altered perception of a face after exposure to a preceding face. In the study by Jeffery et al. participants were adapted to faces, which had been distorted by being contracted to the center of the face. In a next step participants were tested with faces in either the same or different viewpoints. Adaptation to distorted faces usually results in perceiving the test faces as distorted in the other direction, that is to say, when the adapted face is contracted to the center, the test face will be perceived as being distorted in the opposite way. Jeffery et al. found that children and adults showed similar amounts of transfer across changes in face viewpoint.

The aim of the present study was to investigate viewpoint effects on face recognition during childhood thereby adding to the previous literature in three respects. First, our study included recognition of faces in profile views. As the ability to encode profile views of faces is the only face processing ability undergoing qualitative changes during infancy (*McKone et al., 2012*), this delayed development might still be visible during childhood. Second, in the present study we investigated viewpoint effects and viewpoint change effects separately in two experiments. By viewpoint effects we refer to the effect a certain viewpoint has on memory performance per se. These effects can be assessed when presenting a face

in the same viewpoint in study and test phase (experiment 1). Viewpoint change effects on the other hand refer to the effects tested in paradigms, which include a change of the study phase to the test phase (experiment 2). Separating viewpoint effects from viewpoint change effects, allows to investigate if the pattern of performance for profile, three-quarter and front views as found in adult studies, is the same in school-aged children. Moreover, it allows to disentangle if a decrease in performance in a paradigm including a change of the viewpoint is driven by the change of viewpoint or by effects of the viewpoint per se. Third, we compared the development of viewpoint effects in face recognition to viewpoint effects for another object category (here: cars) to assess the specificity of the viewpoint effects for faces.

# EXPERIMENT 1

## Materials & Methods

### Participants

A total of 120 healthy participants took part in this experiment. Data of four participants had to be excluded because of technical problems (two seven-year-olds and two nine-year-olds), because they were too old (one 11-year-old child) or because participants did not complete the experiment (one five-year-old and one six-year-old child). Data of further 25 children (13 five- to six-year-olds, nine seven- to eight-year-olds, three nine- to 10-year-olds) were excluded because they kept looking away from the screen during the experiment or due to preemptive responses as indicated by reaction times shorter than 200 ms on some of their trials. Including these 25 participants, however, did not change the pattern of results. The remaining sample ($n = 88$) consisted of 19 five- and six-year-olds ($M = 5.47$y, $SD = 0.51$, 10 male), 25 seven- and eight-year-olds ($M = 7.52$y, $SD = 0.51$, 17 male), 29 nine- and 10-year-olds ($M = 9.48$, $SD = 0.51$, 15 male) and 15 adults ($M = 21.92$y, $SD = 5.18$y, five male). Children were recruited via local primary schools and day-care-centers and were rewarded with a little present after participation. Adult participants were recruited by announcements at the university and received course credit as compensation. The study was approved by the ethics committee of the faculty of Psychology of Ruhr-Universität Bochum. Parents gave their written consent for their children's participation and adult participants gave their written consent for their own participation.

### Stimuli

Stimuli consisted of 180 photographs of faces and 180 photographs of cars. Images of faces were taken from the Radboud Faces Database (*Langner et al., 2010*) and the Karolinska Directed Emotional Faces database (*Lundqvist, Flykt & Öhman, 1998*). The 180 photographs of faces depict 60 Caucasian adults (30 female) with neutral facial expression shown from three viewpoints: front view (0°), three-quarter view (45°) and profile view (90°). Photographs of cars were taken from homepages of car manufacturers and include images of 60 different car models in grey or silver shown in front, three-quarter and profile view. Images of faces and cars were scaled to a maximum image height or width of 250 pixels and centered in a 300 × 300 pixels-sized square with a grey background. Image processing was done using Adobe Photoshop CS5 Extended. Examples of the stimuli can be seen in Fig. 1.

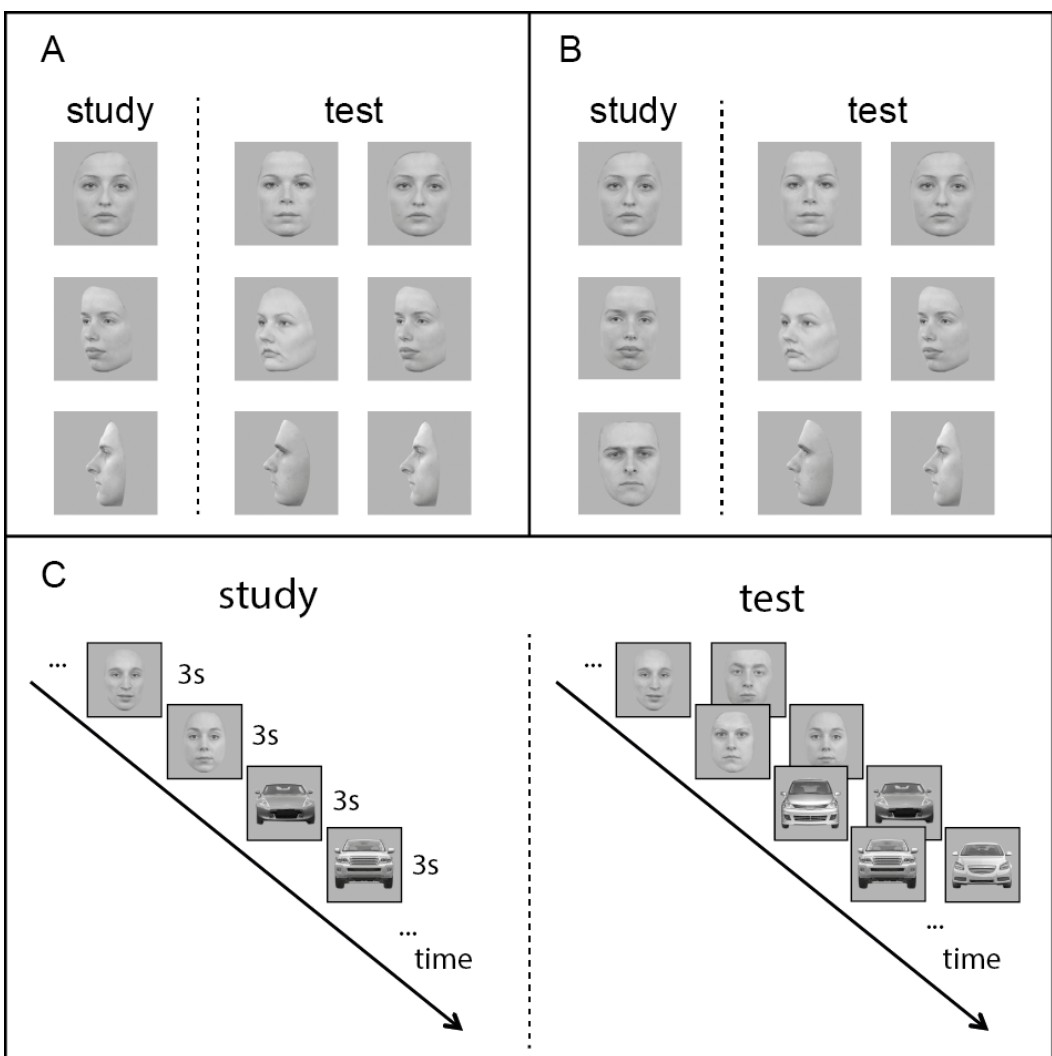

**Figure 1  Stimuli and procedure.** (A) Example face stimuli for the three task versions in experiment 1 (from top to bottom: front, three-quarter, profile). (B) Example face stimuli for the three task versions in experiment 2 (from top to bottom: 0°-, 45°-, 90°-change). (C) General overview on the task procedure.

### *Design and procedure*

The design of the memory task was adapted from *Weigelt et al. (2014)*. Each participant took part in three consecutive versions of this memory task; one version for each of the three viewpoints (front, three-quarter, profile). Each of these tasks consisted of a study and a test phase (see Fig. 1C). In the study phase participants saw 10 images of one category (e.g., faces) followed by 10 images of the other category (e.g., cars). Each image was presented for three seconds and was immediately followed by the next image. The order of stimulus category was counterbalanced between participants and was the same in the test phase, which consisted of a 2AFC task. The test phase was started by button press by the experimenter and was conducted immediately after the study phase. Here, participants had to indicate the previously seen face or car (target) via button press. Whenever an answer

was given, the next pair of images appeared. Face targets were matched with distractor faces of the same sex. Also, no participant saw the same face (or car) in more than one version of the task. We used three different stimulus sets to rule out the influence of certain stimuli on the performance in certain conditions. The use of the stimulus sets in the three conditions was counterbalanced between participants. Experiments were conducted at the university, in primary schools, day-care-centers or at the families' houses. Prior to the actual experiment children took part in a short practice test, using faces of cartoon characters to make sure they had understood the procedure. If children did not answer the practice trials correct, the practice task was repeated. Stimuli were presented on a MacBook Pro with a screen size of $15''$ diagonally using MATLAB (version R2009b, The Mathworks) and the Psychtoolbox (version 3.0.9, *Brainard, 1997*).

### Analysis

Statistical analysis was done using SPSS (version 23). Unless stated otherwise the dependent variable is accuracy in all analyses.

## Results

### Preliminary analyses

*Controlling for the presentation order of stimulus categories.* A repeated-measures ANOVA (rmANOVA) comprising the within-group-factors *category* (faces, cars) and *viewpoint* (front, three-quarter, profile) and the between-group-factors *age group* (5–6, 7–8, 9–10, A) and *category order* (cars-faces, faces-cars) revealed no significant effects of *category order* (all $ps > .05$). Therefore, this factor was excluded from further analyses.

*Testing for floor and ceiling effects.* Floor and ceiling effects are common problems in developmental studies. Here, performance in the youngest age group (five- and six-year-olds) differed significantly from chance (50% correct) in five out of six conditions. The only condition, which was not significantly different from chance, was cars in three-quarter view, $t(18) = 2.072$, $p = .053$. We interpret this finding as evidence that also our youngest age group of children was in general capable of understanding and performing our task. Adults' performance was significantly different from 100% in all conditions (all $ps < .05$).

### Main analyses

*Effects of different viewpoints.* We computed a rmANOVA comprising the within-group-factors *category* (faces, cars) and *viewpoint* (front, three-quarter, profile) and the between-group-factor *age group* (5–6, 7–8, 9–10, A). The rmANOVA revealed the following results: first, a significant main effect of *viewpoint*, $F(2, 168) = 3.405$, $p = .036$, $\eta_p^2 = .039$, demonstrating that memory performance was influenced by the viewpoint stimuli were presented in. Second, a significant main effect of *age group*, $F(3, 84) = 15.404$, $p < .001$, $\eta_p^2 = .355$, reflecting higher memory performance with increasing age of participants. Crucially, the interaction between *viewpoint* and *age group* was not significant, $F(6, 168) = 0.922$, $p = .481$, thereby indicating similar viewpoint effects across age groups (see Fig. 2). Further significant effects were the main effect of *category*, $F(1, 84) = 23.984$, $p < .001$, $\eta_p^2 = .222$, reflecting overall higher performance for faces compared to cars,

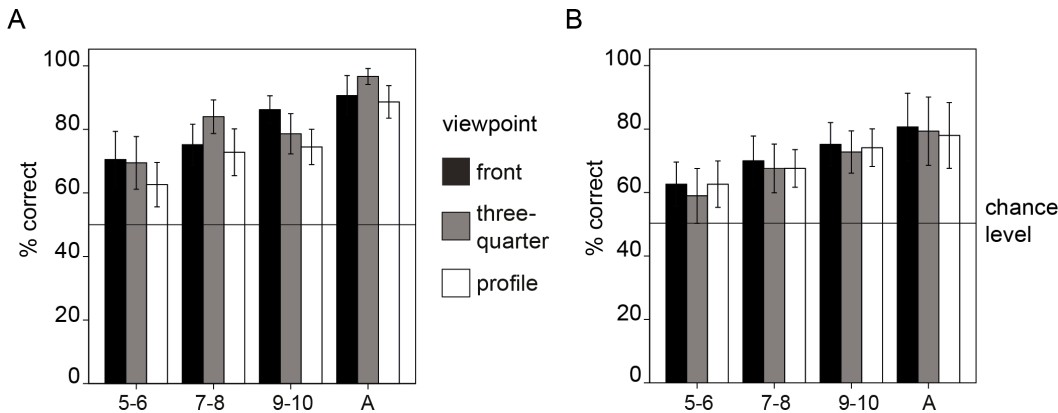

**Figure 2 Results of experiment 1.** Mean performance is shown for faces (A) and cars (B). The age groups are indicated on the *x*-axis and correspond to: five- to six-year-olds, seven- to eight-year-olds, nine-to-10-year-olds and adults. Error bars: ±2 SE.

and the *viewpoint × category* interaction, $F(2, 168) = 4.084$, $p = .019$, $\eta_p^2 = .046$, which showed that viewpoint effects differed for faces and cars. There were no other significant effects (*category × age group*, $F(3, 84) = 0.775$, $p = .511$; *viewpoint × category × age group*, $F(6, 168) = 1.019$, $p = .415$).

To further examine the *viewpoint* effect and the interaction between *viewpoint* and *category*, we computed separate rmANOVAS (with the factors *age group* and *viewpoint*) on the face and car data. In the car data, the effect of age group was significant, $F(3, 84) = 5.276$, $p = .002$, $\eta_p^2 = .159$; however, the effect of *viewpoint* was not significant, $F(2, 168) = 0.7$, $p = .498$, showing that the viewpoint effect in the rmANOVA including both categories was mainly driven by the viewpoint effect in faces. There were no other significant effects (*viewpoint × age group*, $F(6, 168) = 0.118$, $p = .994$).

In the face data, there was also a significant effect of *age group*, $F(3, 84) = 18.578$, $p < .001$, $\eta_p^2 = .399$, and a significant effect of *viewpoint*, $F(2, 168) = 6.41$, $p = .002$, $\eta_p^2 = .071$, but no significant *viewpoint × age group* interaction, $F(6, 168) = 1.721$, $p = .119$, $\eta_p^2 = .058$. Post hoc *t*-tests on the viewpoint effect in faces revealed that performance for three-quarter-views of faces was significantly higher compared to performance for profile views, $t(87) = 3.480$, $p = .001$, $d = 0.43$. Likewise, performance for front views was significantly higher compared to profile views, $t(87) = 3.185$, $p = .002$, $d = 0.39$. Performance for front and three-quarter-views did not differ significantly, $t(87) = -0.339$, $p = .736$.

## Discussion

Results from experiment 1 suggest that the pattern of face recognition performance for profile, three-quarter and front views of faces is similar in school-age children and in adults. The lack of a significant interaction between viewpoint and age group suggests that the pattern of performance elicited by different viewpoints previously reported in studies in adults, i.e., lower performance for profile view compared to both front and three-quarter views of faces (*Bruce, Valentine & Baddeley, 1987*), appears to be present in school-age children already.

## EXPERIMENT 2

The goal of this experiment was to assess the influence of viewpoint changes on face memory performance and on memory performance for a control category (cars) in children and adults.

### Materials & Methods
#### Participants

Participants consisted of a new group of participants ($n = 40$) that had not taken part in experiment 1. In this experiment we decided to test only the youngest age group comprising five- to six-year-olds, because if we were to see developmental changes, we would expect them in the youngest age group. Data of further 22 participants was excluded due to technical problems (one adult, one child), because participants did not want to finish all three tasks (three children) and participants were not able to perform the task (e.g., kept pressing only one button, did not look at the screen or gave preemptive responses as indicated by reaction times shorter than 200 ms, 17 children). The remaining sample consisted of 18 five- to six-year-old children ($M = 5.53$y, $SD = 0.51$, five male) and 22 adults ($M = 21.41$y, $SD = 2.89$, two male). Recruitment and compensation for the experiment was similar to experiment 1.

#### Stimuli and procedure

Stimuli and procedure were identical to experiment 1.

#### Design

We used the same memory task as in experiment 1, however in this experiment we included two conditions with a viewpoint change: In one condition, participants studied items in front view and were tested in three-quarter-view, resulting in a 45° -change between study and test view (45°-condition). In the other condition, participants studied items in front view and were tested in profile view, resulting in a 90° -change between study and test view (90°-condition). Furthermore, participants completed one version of the memory task, in which—as in experiment 1—faces and cars were studied in front view and tested in front as well (0°-condition), to allow for a direct comparison between conditions with and without viewpoint changes. The order of the three conditions (0°, 45°, 90°) was randomized between participants. An overview of all three memory task versions can be seen in Fig. 1B.

### Results
#### Preliminary analyses

*Controlling for the presentation order of stimulus categories.* A rmANOVA including the within-subjects-factors *category order* (faces-cars, cars-faces), *category* (faces, cars) and *viewpoint-change* (0°, 45°, 90°) and the between-subjects-factor *age group* (children, adults) revealed no significant effects of *category order* (all $p > .05$). Therefore, this factor was excluded from further analyses.

*Testing for floor and ceiling effects.* As in experiment 1, children's performance differed significantly from chance in the 0° -viewpoint-change condition for both faces,

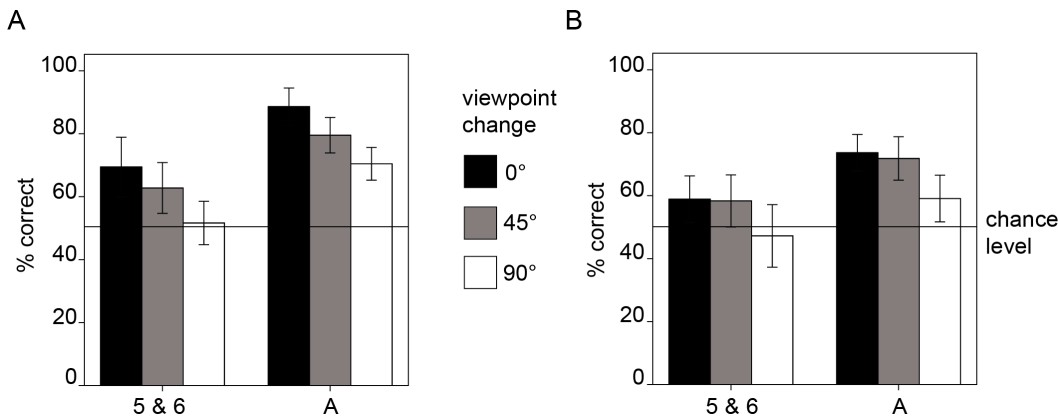

**Figure 3** **Results of experiment 2 for faces (A) and cars (B).** The age groups are indicated on the *x*-axis and correspond to: five-and six-year-olds and adults. Error bars: ±2 SE.

$t(17) = 4.096$, $p = .001$, $d = 0.97$, and cars, $t(17) = 2.406$, $p = .028$, $d = 0.57$. Furthermore, the 45°-condition was significantly different from chance for faces, $t(17) = 3.173$, $p = .006$, $d = 0.75$, and marginally significant for cars, $t(17) = 2.012$, $p = .060$, $d = 0.47$. The 90°-condition was at chance for both faces, $t(17) = 0.483$, $p = .636$, and cars $t(17) = -.559$, $p = .584$. We conclude that, although children's performance was at chance in some conditions, in general children were able to perform the task. Adult data was free from floor or ceiling effects in all conditions (all *ps* < .05).

*Comparison of results from experiment 1 and 2.* As the 0°-condition in experiment 2 is equivalent to the front-condition in experiment 1, we tested in a first step, if results in five-to six-year-olds and adults was similar across experiments. There were no differences in either adults or five- to six-year-olds in performance for front views of faces or front views of cars from experiment 1 to experiment 2 (all *ps* >.05), showing that performance was similar for the groups in experiment 1 and 2.

## Main analyses
### Effects of viewpoint changes
We computed a rmANOVA including the within-group-factors *category* (faces, cars), *viewpoint-change* (0°, 45°, 90°) and the between-group-factor *age group* (children, adults), which revealed the following findings: first, a main effect of *viewpoint change*, $F(2, 76) = 20.328$, $p < .001$, $\eta_p^2 = .349$, showing that memory performance for faces and cars was influenced by changes in viewpoint. Second, a main effect of *age group*, $F(1, 38) = 36.032$, $p < .001$, $\eta_p^2 = .487$, indicating overall better performance for adults than children. Importantly, the interaction between *age group* and *viewpoint change* was not significant, $F(2, 76) = 0.81$, $p = .922$, indicating that memory performance was similarly affected by viewpoint changes in five- to six-year-old children and adults (see Fig. 3). Furthermore, results revealed a main effect of *category*, $F(1, 38) = 17.327$, $p < .001$, $\eta_p^2 = .313$, indicating overall higher performance for faces compared to cars. No other interactions were significant (*category* × *age group*, $F(1, 38) = 1.297$, $p = .262$, $\eta_p^2 = .033$;

*category* $\times$ *viewpoint change*, $F(2,76) = 1.267$, $p = .287$; *category* $\times$ *viewpoint change* $\times$ *age group*, $F(2,76) = .091$, $p = .913$).

To further examine the nature of the *viewpoint change* effect, we computed post hoc $t$-tests to compare the three conditions (0°, 45°, 90°). Memory performance in the 0°-condition differed significantly from the 90°-condition, $t(39) = 6.413$, $p < .001$, $d = 1.13$, and performance in the 45°-condition differed significantly from the 90°-condition, $t(39) = 5.013$, $p < .001$, $d = 0.54$. The difference in performance between the 0°-condition and the 45°-condition was not significant, $t(39) = 1.691$, $p = .099$.

## Discussion

Results from experiment 2 show that the effects of viewpoint changes are similar in five- to six-year-old children and adults. These findings are in line with previous results showing that eight-year-old children and adults were similarly affected by a rotation between front and three-quarter views of faces (*Crookes & Robbins, 2014*). Furthermore, our results extend these findings by showing that also a viewpoint change involving profile views similarly affects face recognition five- to six-year-old children and adults. A limitation to the present findings is that five- and six-year-old children's performance was at chance for the 90°-condition, which might have masked a stronger decline in performance for this group compared to the adult group. Moreover, the exclusion rate in this experiment was higher compared to experiment 1. The reason for this high exclusion rate might be found in the higher difficulty in task 2 compared to task 1 which is also reflected in the performance drop in adults for the conditions involving a viewpoint change. For instance, adults have an accuracy of 89% in the most difficult face-condition in experiment 1 (faces in profile view), but performance drops to about 70% for the 90°-change condition in experiment 2. This difficulty might have influenced children's motivation to properly complete all three tasks. Nevertheless, comparable levels of performance in the 0°-change condition in experiment 2 with the same condition in experiment 1 (front-condition), suggest that the data was not negatively influenced by the exclusion rate.

## GENERAL DISCUSSION

The present study investigated the effects of different viewpoints and viewpoint changes on face recognition performance in school-age-children and adults. The main finding of this study is that face recognition in children and adults is similarly affected by viewpoint effects (experiment 1) and also by viewpoint change effects (experiment 2). Thereby, the present findings are consistent with recent findings suggesting no developmental differences of view-invariant face recognition (*Jeffery et al., 2013*; *Crookes & Robbins, 2014*). Our results extend previous findings by showing that not only the general difference between encoding in the same and across views, but also the pattern of how recognition performance is affected by different viewpoints (experiment 1) and viewpoint changes (experiment 2) appears to be similar between school-age children and adults. Furthermore, our results show that also when including profile views of faces, viewpoint effects and viewpoint change effects are similar in children and adults.

In experiment 1 we assessed the pattern of recognition performance for each of the three viewpoints front, three-quarter and profile. Besides an age-related increase in performance, we found that recognition of faces, but not of cars was influenced by the viewpoint the stimulus was presented in. For the face data, performance in five- to 10-year-old children and adults performance for both front and three-quarter views was significantly higher compared to that for profile views, a pattern that has been found in numerous studies in adults (e.g., *Bruce, Valentine & Baddeley, 1987*; *McKone, 2008*). Also with regard to developmental aspects, profile views can be assigned a special role, as the ability to encode faces in profile view—in contrast to encoding of front and three-quarter views—only emerges at seven months (*Fagan III, 1976*) and therefore seems to be one of the few face processing abilities undergoing qualitative changes. However, the present results suggest that after that qualitative change has taken place, the pattern of performance for profile, three-quarter and front views seems to be quite stable at least from child- to adulthood.

In contrast to faces, for cars, we did not observe viewpoint effects. One factor that seems to be relevant for performance differences between several viewpoints of objects is the familiarity with certain viewpoints (see, *Blanz et al., 1999*). Therefore, a possible explanation for the absence of viewpoint effects in cars in the present study would be that in normal life we see cars roughly equally often from all three tested views.

In experiment 2 we investigated the effects of viewpoint changes on recognition performance. We found an overall increase of performance with age, a significant effect of viewpoint change but no interaction between viewpoint change and age, showing that viewpoint change effects were similar across the age groups. In the face data, all three conditions differed significantly from each other with the performance being highest for the 0°-condition, lower for the 45°-condition and with lowest performance for the 90°-condition. This finding matches results from previous studies in adults reporting that when participants learned front views of faces, performance decreased with increasing angle between study and test phase (*Hill, Schyns & Akamatsu, 1997*). Also, results from experiment 2 show that the decrease of performance in a condition containing a viewpoint change (45°- and 90°-condition) compared to a condition without change (0°-condition) is similar in children and adults, indicating that five-and six-year-old children's representations of faces are not more view-specific than those of adults. Thereby our results are consistent with a study by *Crookes & Robbins (2014)*, who found that eight-year-old children and adults were similarly affected by a viewpoint change (study front and test three-quarter) compared to a condition without viewpoint change (study front and test front). The present findings conflict with results from studies suggesting that the ability to recognize faces across different views matures late (*Mondloch et al., 2003*). A possible explanation for the divergence of results is that the study by *Mondloch et al. (2003)* also included up and down rotations of the face and not only changes between front and three-quarter views.

A limitation to the present study is that overall performance was lower for cars compared to faces, indicating that the car stimuli were more difficult, probably due to the selection of images of very similar car models. Ideally, overall performance levels for faces and cars would have been matched. Furthermore, in experiment 2 children were at chance for both

faces and cars in the 90°-condition, showing that this condition was too difficult for this age group. However, since performance was not at chance in the 45°-condition we can conclude that in general also the youngest age group was able to perform the task and that the low performance in the 90°-condition is indeed informative.

## CONCLUSION

In conclusion, the present study shows similar viewpoint and viewpoint-change-effects on face-memory-performance in school-age children and adults. Our results suggest that the pattern of performance for profile, three-quarter and front views of faces seems to be quite stable at least from child- to adulthood and that face recognition is similarly affected by changes across these views in children and adults. In sum, our results suggest early maturity of the ability to recognize faces in and across different viewpoints.

## ACKNOWLEDGEMENTS

We thank Astrid Hönekopp, Tobias Meißner, Luzie Mount, Helen Prüfer, Rebecka Röhnke, Katharina Sommer, Sophia Terwiel and Ricarda F. Weiland for their assistance in data collection.

### Funding

This work was supported by a scholarship of the German National Academic Foundation (Studienstiftung des deutschen Volkes) and grants from the German Research Foundation (WE 5802/1-1) and the Daimler and Benz Foundation. The funders had no role in study design, data collection and analysis, decision to publish, or preparation of the manuscript.

### Grant Disclosures

The following grant information was disclosed by the authors:
German National Academic Foundation.
German Research Foundation: WE 5802/1-1.
Daimler and Benz Foundation.

### Competing Interests

The authors declare there are no competing interests.

### Author Contributions

- Marisa Nordt performed the experiments, analyzed the data, wrote the paper, prepared figures and/or tables.
- Sarah Weigelt conceived and designed the experiments, contributed reagents/materials/analysis tools, reviewed drafts of the paper.

### Human Ethics

The following information was supplied relating to ethical approvals (i.e., approving body and any reference numbers):

The ethics board of the faculty of Pschology of Ruhr-Universität Bochum granted Ethical approval to carry out the study within its facilities (Application nr. 104).

## Data Availability

Data can be found at the Open Science Framework:

https://osf.io/gqvwb/.

## Supplemental Information

Supplemental information for this article can be found online at http://dx.doi.org/10.7717/peerj.3253#supplemental-information.

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
