# Peer review of "Face recognition is similarly affected by viewpoint in school-aged children and adults"

_PeerJ, doi:10.7717/peerj.3253_

## Round 0.1 · original submission · Minor Revisions

· Academic Editor

Minor Revisions

Dear Authors, 3 peer reviewers have given excellent suggestions how to improve your manuscript which will make your publication highly citable. Please do the needed revisions and resubmit the revised manuscript as soon as possible.

·

Basic reporting

The paper is organized and well written. The topic is of current importance. The conclusions are fairly justified by the data. Only some minor issues and suggestions stated in these 3 sections:
The Introduction could be better streamlined by adding some clear hypotheses/predictions.

Experimental design

1. The details of the testing phase of both experiments should be clarified. For example, it is not clear to me how long was the presentation of each stimulus, and how many trials were per condition.
2. Same for the analysis method: Were the mean reaction time (RT) used in the statistical analyses calculated for each condition and for each participants from all trials or from only correct trials? What were the error rates of the individual performance? Was there any outliers, e.g., the individual reaction time exceeds 1.96 standard deviations below or above participant’s own mean?
3. In the statistical analyses, did the authors do the normal distribution test on the data? Or was any sorts of assessment of sphericity and homogeneity of variance for each ANOVA done?

Validity of the findings

I struggled a bit to understand how did the authors reach the conclusion of the early maturation of face recognition ability across different viewpoints when the performance of young children in recognizing profile faces was at floor level whereas that of adults was clearly above chance. But I can see where the claims are coming from, as the difference across conditions found in experiment 2 may falsely appear to be same but it is very unlikely that children at young age would outperform adults in profile face recognition with a task that is not too difficult for the young children nor too easy for the adults. However, I wonder could this result be also interpreted as children have difficulties in generalizing learnt configural face information to recognize faces in the profile view, which in turn suggests qualitatively different face configural processing in children compared to adults?

Reviewer 2 ·

Basic reporting

Good.

Experimental design

Methodological information gap. Need to improve.

Validity of the findings

No comments.

Comments for the author

The manuscript ‘Face recognition is similarly affected by viewpoint in school-aged children and adults’ is an interesting manuscript and has valuable information. However I have few issues on it.


1. In abstract, line 3 from above, better to give age range of both children and adult as this line is not clear to understand. Probably better to delete the word ‘including’. Please mention the participants’ number for experiment 1 and 2 separately in abstract.
2. Please state in abstract and in methodology section that how author/authors collect or measure the performance of participants for both groups.
3. It is not clear the importance of this study. Please mention the importance in ‘Introduction’ section.
4. In Introduction, please mention ‘year’ or ‘month’ after value of age (unit missing). These were observed throughout the ‘Introduction’.
5. Page 6, in abstract author/authors mentioned the number of participants is 128. Here they explained the exclusion of 7 subjects. Please revise carefully and explain the one more subject exclusion.
6. Page 8, last line is not clear. “................. presented for three seconds each”. Is each image presented for three seconds? And what is the inter stimulus interval (ISI)? Please mention here.
7. Page 9 and 13, the ‘result’section looks analysis and explanation of results in together. Therefore headline should be changed to “Data analysis and results”.
8. Results should be expressed according to the standard practice, that is, something like “F(2,27) = 4.467, p = .021” (three digit later). The same across all the text.
9. Page 12, line 260. Author/authors mentioned “n=40”. Is this number for each group separately or combine for both groups? Confused. Please make it clear with matching the information of line 267 and 268.
10. As there is no ‘special discussion’ in this manuscript, therefore only “Discussion” headline would be better instead of ‘general discussion’.

·

Basic reporting

The paper looks interesting and is mostly well written. The relevant literature is adequately covered. There are a still a couple of points the authors should address:

Abstract (p. 2): The abstract only refers to the results obtained for faces but fails to mention the control condition (car pictures). Please add.

Page 2, line 57 (“…improvement of face recognition abilities between six and 10 years, followed by a short plateau)”: This observation is attributed to Lawrence et al. (2008). However, the existence of such a plateau has been reported much earlier by Carey, Diamonds and Woods (1980). The authors should acknowledge that.

P. 2, l. 68 (“The counterpart is the early maturity hypothesis…”): Odd wording, I suggest something along the lines of “As an alternative explanation, the early maturity hypothesis has been proposed. It states…”

P. 3, l. 85-99 (Turati et al., Nakato et al.): Account of Turati et al’s work rather unclear. It should be pointed out that the preference test in that study involved the pairing of the target face with a novel face, otherwise the term “novelty preference” remains meaningless. Similarly, the description of Nakato et al. needs more detail. For example, what are changing oxy-Hb levels indicative of? etc. Please explain.

P. 5, l. 120 (“face aftereffects”): Briefly explain.

P. 6, l. 123 (“…in three aspects”): Should read “…respects”.

P. 6, l. 126-139: Change present tense to past tense (l. 124 “included”, l. 127 “investigated”, l. 137 “compared”).

P. 11, l. 245: When reporting several experiments in a paper it is good practice to add, at the end of the result sections of each experiment (Exp. 1: l. 245, Exp. 2: l. 327), a short discussion of the main findings, before moving on to the General discussion.

Otherwise, the structure conforms to PeerJ standards, the figures are ok and the raw data has been supplied.

Experimental design

The reported research falls within the scope of PeerJ. The research question is well defined. The study appears well motivated, and has been carefully conducted. All ethical guidelines have been followed.
The method is described in sufficient detail, except for the following: Fig. 1 illustrates study and test phase but does not specify how long these two phases were separated in time? Nor is this information provided in the main text. Please specify.

Validity of the findings

The data analysis generally looks fine, except for the following points which the authors should address:
- Exclusion of participant data: In Experiment 2 it seems that about half of the participating children were excluded, which seems quite a high rate. Does this affect the validity (generalisation) of the findings? Please comment.
Related to that it is unclear whether the statement n=40 (line 260) refers to each age group or to both groups combined.
- In the context of the results of Exp. 1, the follow-up analysis for the face data (l. 288-244) only gives the main effects. How about the age x viewpoint interaction?
- I wonder whether it would also be of interest to analyse the asymptotic behaviour of the developmental trajectories shown in Fig. 2. At what age is performance equivalent to that of adults? This might provide cross-links to the early and late maturation hypotheses mentioned in the introduction (p. 3).
- A minor point: When reporting ANOVA results it’s good practice (and facilitates reading) to follow a structured approach, by stating all main effects first, followed by all two-way interactions, three-way interactions, etc. before turning to any post hoc tests carried out to follow up any significant interaction terms.

Discussion mostly ok. Regarding the limitations of the study (p. 17, l. 382) one could argue that the absence of an age x viewpoint change interaction in Exp. 2 (Fig. 3) could be due to the fact that children were performing at chance level for the 90 deg view changes and that this may have masked a potentially larger performance decline in that age group. The authors should address this possibility in their discussion.

---

## Round 0.2 · Minor Revisions

· Academic Editor

Minor Revisions

Dear Authors,

Please correct the minor mistakes mentioned by one of the peer reviewers. Rather than correct this post-Acceptance, it is best that you check this doesn't influence anything else in the text, and return a submission which we will then promptly Accept.

·

Basic reporting

No specific comments.
The revised manuscript reaches criteria for this section.

Experimental design

No specific comments.
The revised manuscript reaches criteria for this section.

Validity of the findings

No specific comments.
The revised manuscript reaches criteria for this section.

Comments for the author

Please delete 'in experiment' in line 365 '... but in experiment 1 performance drops to about 70% for the 90-change condition in experiment 2.'

Reviewer 2 ·

Basic reporting

Good.

Experimental design

Good.

Validity of the findings

Ok.

Comments for the author

The revised manuscript is now well written.

·

Basic reporting

The reporting looks fine now. Almost all issues raised in my previous review have been adequately addressed. Just one small point:
Page 3, line 58 (“…short plateau, but then again found further face recognition improvement starting from age 13 (Lawrence et al., 2008) or 16 (Carey, Diamond & Woods, 1980)”:
This is still not quite correct: Carey et al. reported the plateau at an age range of around 10-13 (rather than 16) – see Figures 2 and 3 in their paper.

Experimental design

This looks fine now.

Validity of the findings

The data analysis looks fine, all issues raised in my previous review have been adequately addressed.

---

## Round 0.3 · accepted · Accept

· Academic Editor

Accept

Dear Authors,

The manuscript has been revised satisfactorily and will be accepted for publication in PeerJ.